# Post-Concussion Symptoms in Complicated vs. Uncomplicated Mild Traumatic Brain Injury Patients at Three and Six Months Post-Injury: Results from the CENTER-TBI Study

**DOI:** 10.3390/jcm8111921

**Published:** 2019-11-08

**Authors:** Daphne C. Voormolen, Juanita A. Haagsma, Suzanne Polinder, Andrew I.R. Maas, Ewout W. Steyerberg, Petar Vuleković, Charlie A. Sewalt, Benjamin Y. Gravesteijn, Amra Covic, Nada Andelic, Anne Marie Plass, Nicole von Steinbuechel

**Affiliations:** 1Department of Public Health, Erasmus MC, University Medical Center Rotterdam, PO Box 2040, 3000 CA Rotterdam, The Netherlands; j.haagsma@erasmusmc.nl (J.A.H.); s.polinder@erasmusmc.nl (S.P.); e.steyerberg@erasmusmc.nl (E.W.S.); c.sewalt@erasmusmc.nl (C.A.S.); b.gravesteijn@erasmusmc.nl (B.Y.G.); 2Department of Emergency Medicine, Erasmus MC, University Medical Center Rotterdam, 3000 CA Rotterdam, The Netherlands; 3Department of Neurosurgery, Antwerp University Hospital, 2650 Edegem, Belgium; Andrew.Maas@uza.be; 4Department of Neurosurgery, University of Antwerp, 2000 Edegem, Belgium; 5Department of Medical Statistics and Bioinformatics, Leiden University Medical Center, 2333 ZA Leiden, The Netherlands; 6Clinic of Neurosurgery, Clinical Centre of Vojvodina, 21000 Novi Sad, Serbia; pvulekovic@gmail.com; 7Institute of Medical Psychology and Medical Sociology, University Medical Göttingen (UMG), 37075 Göttingen, Germany; amra.covic@med.uni-goettingen.de (A.C.); annemarie.plass@med.uni-goettingen.de (A.M.P.); nvsteinbuechel@med.uni-goettingen.de (N.v.S.); 8Department of Physical Medicine and Rehabilitation, Oslo University Hospital, 0372 Oslo, Norway; nandelic@online.no; 9Faculty of Medicine, Institute of Health and Society, Research Centre for Habilitation and Rehabilitation Models and Services (CHARM), University of Oslo, 0318 Oslo, Norway

**Keywords:** complicated mild traumatic brain injury, post-concussion symptoms, post-concussion syndrome, traumatic brain injury

## Abstract

The aim of this study was to assess the occurrence of post-concussion symptoms and post-concussion syndrome (PCS) in a large cohort of patients after complicated and uncomplicated mild traumatic brain injury (mTBI) at three and six months post-injury. Patients were included through the prospective cohort study: Collaborative European NeuroTrauma Effectiveness Research (CENTER-TBI). Patients enrolled with mTBI (Glasgow Coma Scale 13–15) were further differentiated into complicated and uncomplicated mTBI based on the presence or absence of computed tomography abnormalities, respectively. The Rivermead Post-Concussion Symptoms Questionnaire (RPQ) assessed post-concussion symptoms and PCS according to the mapped ICD-10 classification method. The occurrence of post-concussion symptoms and syndrome at both time points was calculated. Chi square tests were used to test for differences between and within groups. Logistic regression was performed to analyse the association between complicated versus uncomplicated mTBI and the prevalence of PCS. Patients after complicated mTBI reported slightly more post-concussion symptoms compared to those after uncomplicated mTBI. A higher percentage of patients after complicated mTBI were classified as having PCS at three (complicated: 46% vs. uncomplicated: 35%) and six months (complicated: 43% vs. uncomplicated 34%). After adjusting for baseline covariates, the effect of complicated versus uncomplicated mTBI at three months appeared minimal: odds ratio 1.25 (95% confidence interval: 0.95–1.66). Although patients after complicated mTBI report slightly more post-concussion symptoms and show higher PCS rates compared to those after uncomplicated mTBI at three and six months, complicated mTBI was only found a weak indicator for these problems.

## 1. Introduction

In the European Union, around 2.5 million new cases of traumatic brain injury (TBI) occur each year [1]. The vast majority of patients presenting to hospital with a TBI are diagnosed as having mild TBI (mTBI; Glasgow Coma Score (GCS): 13–15) [1]. Some of these patients may have traumatic intracranial abnormalities on the computed tomography (CT) performed on presentation which could potentially be associated with worse outcomes compared to those who do not have any traumatic intracranial abnormalities. For this reason, Williams et al. [2] addressed a subgroup conceptualization of these injuries, which has been shown to provide more detail on level of outcome [3]. According to this approach a differentiation can be made between patients with a complicated (intracranial abnormalities present on CT) and uncomplicated (no intracranial abnormalities present on CT) mTBI. 

In addition to heterogeneity with regard to the manifestation of mTBI, outcome may also vary between patients. Many patients with mTBI experience post-concussion symptoms in the first couple of weeks and months following the brain injury. However, the type, amount and severity of these symptoms differ between patients and may fluctuate over time [4]. These post-concussion symptoms could be physical, cognitive, emotional and/or behavioural [5]. When a patient experiences a certain combination of symptoms for longer than three months, they may be diagnosed with post-concussion syndrome (PCS) [6,7]. Generally, the International Classification of Diseases (ICD)-10 [8], or Diagnostic and Statistical Manual of Mental Disorders (DSM)-IV [9] criteria are used to diagnose PCS [6,10]. PCS has been a critically debated topic and the question has been raised if we can, or even should, identify this as a unique syndrome for TBI [4,5,11,12], however, the concept is still used in the majority of post-concussion symptom research.

A certain percentage of patients (estimated between 5% and 43% [13,14,15,16,17,18]) after mTBI report and experience post-concussion symptoms for months and sometimes even longer post-injury [3,19]. However, the literature concerning similarities and dissimilarities of post-concussion symptoms between complicated mTBI and uncomplicated mTBI is inconclusive. McMahon et al. noted that patients after complicated mTBI reported significantly more post-concussion symptoms compared to patients after uncomplicated mTBI at both six and twelve months [20]. On the contrary, Iverson et al. have determined that patients after complicated mTBI reported fewer depression and post-concussion symptoms compared to patients after uncomplicated mTBI [21]. When considering PCS, McCauley et al. demonstrated that abnormalities on CT were not associated with PCS at 3 months following injury [22]. Additionally, Iverson et al. have reported: ‘no significant difference in the percentages of patients in the uncomplicated versus complicated mTBI groups who met ICD-10 criteria for PCS’ [21]. Furthermore, in previous research, limited information has been documented on the different care paths patients after complicated and uncomplicated mTBI may have followed and if reporting of post-concussion symptoms differs between these care paths. 

Despite a growing body of literature on complicated versus uncomplicated mTBI, to date, most studies that compared self-reported symptoms following complicated and uncomplicated mTBI were limited in sample size, and there is a relative paucity of recent data. 

We hypothesize that patients after complicated mTBIs report more post-concussion symptoms and have higher prevalence rates of PCS at both time points compared to those with uncomplicated mTBI. Additionally, we anticipated a larger number of patients after complicated mTBI admitted to hospital ward compared to those discharged home from the emergency room (ER) stratum, and aimed to explore if such patients may constitute an “enriched” population in terms of occurrence of PCS and a higher number of post-concussion symptoms. This would be particularly relevant when planning a clinical trial investigating efficacy of approaches to treat PCS symptoms.

The objectives of this study were to assess the occurrence of post-concussion symptoms and PCS in a large sample of patients after complicated and uncomplicated mTBI at three and six months post-injury. 

## 2. Methods

### 2.1. Study Design 

Patients were included in the Collaborative European NeuroTrauma Effectiveness Research (CENTER-TBI) research project, which is a multicentre, prospective observational longitudinal cohort study, conducted in Europe and Israel [1,23]. The core study enrolled patients with all severities of TBI who presented to centres between 19 December 2014 and 17 December 2017. Inclusion criteria were a clinical diagnosis of TBI, an indication for CT scanning, presenting to a centre within 24 h of injury, and obtained informed consent adhering local and national requirements: prior to inclusion, either personally, or through a legally designated representative [23]. Participants were free to withdraw at any point in time during the study without stating a reason [23]. Patients were excluded when there was a severe pre-existing neurological disorder, i.e., cerebrovascular accident, transient ischemic attacks, and epilepsy, which could potentially invalidate outcome assessments. Three strata were used to prospectively differentiate patients by care path: ER (patients evaluated in the ER and discharged afterwards), admission (patients admitted to hospital ward) and intensive care unit (ICU) (patients who were primarily admitted to the ICU) [23]. The main descriptive findings of CENTER-TBI have been published [24].

### 2.2. Study Participants 

In the current study, only patients with a mTBI diagnosis were included (Glasgow Coma Scale (GCS) 13-15). They were divided in complicated mTBI, which was defined as GCS 13–15 and presence of any intracranial injury on first CT and uncomplicated mTBI, which was defined as GCS 13–15 and absence of any intracranial injury on first CT. For analyses of post-concussion symptoms including PCS, we performed a complete case analysis, selecting all patients after mTBI who completed the Rivermead Post-Concussion Symptoms Questionnaire (RPQ) [25] at three and six months follow-up (*N* = 1302) (Figure 1). 

### 2.3. Measurement-Instrument 

Post-concussion symptoms were assessed by the RPQ [25], which evaluates the frequency and severity of 16 post-concussion symptoms. Symptoms evaluated included headaches, dizziness, nausea/vomiting, noise sensitivity, sleep disturbance, fatigue, being irritable, feeling depressed or tearful, feeling frustrated or impatient, forgetfulness, poor concentration, taking longer to think, blurred vision, light sensitivity, double vision, and restlessness. Patients rated the severity of the post-concussion symptoms on a five-point Likert scale, where 0 represents a rating corresponding to “not experienced at all”, 1 “no more of a problem than before the TBI”, 2 “a mild problem”, 3 “a moderate problem”, and 4 “a severe problem” [25]. To obtain the total score, the ratings of all 16 items are summated, excluding the ratings of 1 [25]. The RPQ was administered at three and six months following injury, and patients were asked to rate the severity of the symptoms over the last 24 h.

Various approaches to defining PCS exist [26]. For this study, we primarily focused on the mapped ICD-10 classification method, in which we defined patients as having PCS when they reported any three of the seven symptoms described in the ICD-10 criteria (e.g., headache, dizziness, sleep disturbance, fatigue, being irritable/easily angered, forgetfulness/poor memory, and poor concentration) [8]. As previous research is inconclusive concerning which severity rating should be applied as a cut-off [26], two different cut-offs were assessed: rating score 2 (≥2), corresponding to symptoms rated as mild or worse, and rating score 3 (≥3), corresponding to symptoms rated as moderate or worse. 

The RPQ was collected by telephone and face-to-face interviews, or per postal or web-based questionnaires (Appendix A). The questionnaire was translated into 18 languages and linguistically validated [23,27]. 

### 2.4. Ethical Approval 

The CENTER-TBI study (EC grant 602150) has been conducted in accordance with all relevant laws of the EU if directly applicable or of direct effect, and all relevant laws of the country where the Recruiting sites were located, including, but not limited to, the relevant privacy and data protection laws and regulations (the “Privacy Law”), the relevant laws and regulations on the use of human materials, and all relevant guidance relating to clinical studies from time to time in force including, but not limited to, the ICH Harmonised Tripartite Guideline for Good Clinical Practice (CPMP/ICH/135/95) (“ICH GCP”) and the World Medical Association Declaration of Helsinki entitled “Ethical Principles for Medical Research Involving Human Subjects”. Ethical approval was obtained for each recruiting site. Informed Consent was obtained for all patients recruited in the Core Dataset of CENTER-TBI and documented in the e-CRF. The list of sites, Ethical Committees, approval numbers, and approval dates can be found on the official Center TBI website (www.center-tbi.eu/project/ethical-approval).

### 2.5. Statistical Analysis

For all analyses, data was extracted from the INCF Neurobot tool (INCF, Sweden), a clinical study data management tool. Version 2.0 of the CENTER-TBI dataset (data frozen in January 2019) was used in this manuscript. The number and percentage of patients who were classified as having experienced mTBI (complicated and uncomplicated mTBI) were assessed by stratum and per GCS level. Descriptive analyses for demographic data (age, gender and education) injury mechanism, GCS at baseline, first CT scan, and RPQ total score were performed and examined for patients with mTBI, complicated and uncomplicated mTBI at 3 and 6 months post-injury. Chi-square tests for categorical variables and Student’s t tests for continuous variables were used to compare patients with mTBI who completed the RPQ (filled in all items) with those with incomplete RPQ data (not all items filled in). Additionally, these tests were also performed to compare patients with complicated versus uncomplicated mTBI. To assess whether there was a relation between having a completed RPQ at three and/or six months, a McNemar test was performed. 

At each time point, we computed the prevalence and percentages of post-concussion symptoms and of patients who were classified as having PCS according to our mapped ICD-10 classification method, and we explored whether there were differences between the complicated and uncomplicated group. For all analyses, a *p*-value of *p* < 0.05 was considered significant. 

For the analysis of the effect of complicated versus uncomplicated mTBI on PCS, data for the following predictor variables: age, GCS, stratum, education, gender, psychiatric medical history, and cause of injury were first multiply imputed. We assumed missing at random as the mechanism of missingness. For the component variables considering psychiatric medical history (anxiety, depression, sleep disorders, schizophrenia, substance abuse disorder, and other), missings were treated as absence of this diagnosis, since investigators could only enter components if the main category (e.g., psychiatric medical history) had been scored positive. All potential confounders, which were based on clinical relevance, the outcome (PCS), and exposure (complicated TBI) were included in the imputation model. Only the cases with observed outcomes were analysed in the main analysis. The Multivariate Imputation by Chained Equations (mice) package, which imputes incomplete multivariate data by chained equations [28], was used to create five datasets [29]. Results of each imputed data set were combined according to Rubin’s rules [30].

To analyse the association of complicated versus uncomplicated mTBI on the presence of PCS, logistic regression was performed. We adjusted for the following baseline covariates: age, gender, education, injury mechanism, GCS, complicated vs. uncomplicated, and stratum. The unadjusted and adjusted effects were displayed as odds ratios (OR) with 95% confidence intervals (95% CI).

All statistical analyses were performed using SPSS version 25 for Windows (IBM SPSS Statistics, SPSS Inc, Chicago, IL, USA) and R (version 3.2.2 or higher, the R Foundation for Statistical Computing, Vienna, Austria). 

## 3. Results

### 3.1. Study Population 

Within CENTER-TBI, most patients were classified as mTBI (*N* = 2955; 65.5%), and these patients constituted the vast majority of patients in the ER (97%) and admission (93%) strata, but were also present in the ICU (34%) stratum (Table 1). Complicated mTBI was identified in 12%, 45%, and 73% of mTBI in the ER, admission and ICU strata, respectively. 

Figure 2 shows the total number of patients per stratum and provides additional differentiation by GCS score for complicated and uncomplicated mTBI. A larger number of patients with complicated mTBI were found in the admission and ICU strata compared to the ER stratum. Many patients had a significantly lower GCS score when looking at complicated versus uncomplicated mTBI (*p* < 0.01). 

A total of 1302 patients with mTBI and a completed RPQ from the CENTER-TBI database were included in this study (Figure 1). Table 2 shows the characteristics of our study sample. The median age of patients after mTBI was 53 years (interquartile range (IQR); 35–66) and 64% were male. The median number of years of education was 14 (IQR; 11–17) and almost half (47%) of the patients were injured due to an incidental fall, followed by road traffic incidents (39%). Approximately 46% showed any intracranial injury on the first CT and was defined as complicated mTBI. 

Patients after complicated mTBI were significantly older (*p* < 0.01) and had a higher total RPQ score (*p* < 0.01: 11.8 vs. 9.4) compared to patients after uncomplicated mTBI. Patients after mTBI who completed the RPQ were not significantly different from those with incomplete RPQ data, except that they had a slightly higher number of education years (*p* < 0.01: 13.9 vs. 12.6) and more patients reported to have had a psychiatric medical history (*p* < 0.01) (Appendix A). Additionally, there was no statistically significant difference between patients who had a completed RPQ at three and/or six months (*p* = 0.17). 

### 3.2. Post-Concussion Symptoms and PCS

The median RPQ score for patients after complicated mTBI at three and six months was seven (IQR 3 months; 2–20/IQR 6 months; 2–17), which was significantly higher than the median score for patients after uncomplicated mTBI (IRQ: 0–14) (*p* < 0.01). 

Figure 3A shows that patients after complicated mTBI reported significantly more feelings of dizziness, noise sensitivity, fatigue/tiring more easily, feeling depressed/tearful, feeling frustrated or impatient, forgetfulness/poor memory, poor concentration, taking longer to think, and restlessness compared to uncomplicated mTBI at three months (*p* < 0.05).

Additionally, when inspecting data at six months, patients in both groups reported a lower percentage of symptoms and less symptoms were found to be significant, compared to the 3 month time point (Figure 3B). Nevertheless, differences in symptom reporting between complicated and uncomplicated remain. 

PCS prevalence, when using rating score of two as a cut-off, for patients after complicated mTBI were 45.6% (95% CI: 41.6–49.6) and 42.7% (95% CI: 38.7–46.7) at three and six months, respectively, which showed a significant decrease (*p* < 0.01) (Figure 4). A significant difference was also found when comparing the prevalence rates for patients after uncomplicated mTBI, which were 35.3% (95% CI: 31.6–39.0) at 3 months and 34.4% (95% CI: 30.7–38.1) at 6 months. Additionally, significant difference was found between the two groups at both follow-up points (3 months: *p* < 0.01 and 6 months: *p* < 0.01). 

When using rating score of three as a cut-off, post-concussion symptom percentages (Appendix A) and PCS prevalence rates were reduced by half for both complicated and uncomplicated mTBI. Furthermore, there was no significant difference anymore between patients with complicated and uncomplicated mTBI (3 months: *p* = 0.055 and 6 months: *p* = 0.303).

Since the absolute percentages within both groups did not differ much from three to six months, we looked more specifically into the patterns behind the significant difference, which could be explained by the transitioning of patients with PCS between three and six months. Figure 5 shows the trajectories of patients after complicated and uncomplicated mTBI classified with PCS over time. When looking at the complicated mTBI group, there are 272 patients who did not, and 202 patients who did meet the PCS criteria at 3 and 6 months. Seventy-one patients with PCS at 3 month follow-up did not classify as having PCS at 6 month follow-up. Furthermore, 54 patients did not have PCS at 3 months, however, they did classify at 6 month follow-up. In general, the absolute number of patients transitioning between 3 and 6 month follow-up for uncomplicated and complicated mTBI are almost even. 

Figure 6A shows the differentiation per stratum for patients after complicated and uncomplicated mTBI classified as having PCS. The PCS prevalence for patients after complicated mTBI were 43.6% and 51.1% for the admission and ICU strata, respectively. These percentages were higher than the 37.7% admission and 48.3% ICU PCS prevalence rates which were found for patients after uncomplicated mTBI. When looking at 6 months, the reported percentages of patients classified as having PCS for both groups were very similar within the admission stratum, however, the ER and ICU stratum stayed around the same (Figure 6B).

Table 3 shows a summary of a covariate adjusted analysis for the association of complicated and uncomplicated mTBI on the presence of PCS. After adjusting for baseline covariates, the association of complicated versus uncomplicated mTBI at 3 month follow-up was of only borderline significance with an odds ratio (OR) of 1.25 (95% CI: 0.95–1.66). This implies that the difference in PCS prevalence between complicated versus uncomplicated mTBI at 3 months can be influenced by differences in baseline characteristics. However, at 6 month follow-up, the association was rendered insignificant (OR: 1.07, 95% CI: 0.80–1.42) (Appendix A).

## 4. Discussion

This study focussed on the three and six month prevalence rates of post-concussion symptoms and PCS of patients after complicated and uncomplicated mTBI included in a large European database. Overall, we demonstrated that patients after complicated mTBI report significantly more symptoms and have higher prevalence rates of PCS at these points in time. The differences observed at three months (complicated: 45.6% vs. uncomplicated: 35.3%) and six months (complicated: 42.7% vs. uncomplicated 34.4%), were in line with a previous study done by McMahon et al. [20]. However, we found a decrease in symptom reporting from three to six months for both groups, and this is in contrast with McMahon et al. since they determined that patients after uncomplicated mTBI were stable in their symptom reporting across the follow-up times, whereas patients after complicated mTBI reported significantly higher symptoms at six and 12 months compared to three months. Additionally, we found that complicated and uncomplicated mTBI patients transition between being classified with PCS at 3 and 6 month follow-up, which meant that some patients with PCS ‘recover’ after 3 months and some enter the threshold of PCS from 3 to 6 months. Depending on the analysis approach taken, rating score of two or three, we observed variability in results. However, our results confirm a higher prevalence of PCS in patients after complicated compared to uncomplicated mTBI across both approaches. When looking at the association of complicated and uncomplicated mTBI on the presence of PCS, it became less clear after adjusting for baseline covariates, which suggests that the reported differences in this study may be explained by differentiations in baseline characteristics. Lastly, a larger number of complicated mTBI patients were found in the admission and ICU strata, and the percentages of patients classified as having PCS was also higher for both these strata when comparing patients after complicated and uncomplicated mTBI. In terms of PCS occurrence, patients in the admission and ICU strata would appear to represent an “enriched” population, but it is not clear if targeting patients with complicated mTBI would lead to substantial additional enrichment. 

In previous research, contradictions in reporting differences regarding post-concussion symptom and syndrome following complicated and uncomplicated mTBI has led to a conundrum, which is based on the question if patients after complicated mTBI are similar or dissimilar based on symptom reporting compared to uncomplicated mTBI patients. Furthermore, the discussion still stands if post-concussion symptoms are TBI specific [31] and regarding the existence of post-concussion syndrome [4,32,33]. There is no gold standard regarding defining PCS and which severity rating should be used as a cut-off [26], and different approaches to analysis, classification and quantification of PCS exist and lead to variability in results [26]. This has also been demonstrated in this study, where the different severity rating scores have shown to have a substantial impact on the results, since the symptom percentages and PCS prevalence drop down to half when using rating score of three as a cut-off. 

The current study is unique compared to previous studies, because none of them have looked at a large sample such as in this study to compare self-reported symptoms from patients after complicated and uncomplicated mTBI nor did they assess both concepts, post-concussion symptoms and PCS alongside each other. 

A number of limitations of our study should be recognized. No information was available considering if patients were involved in a litigation. Lees-Haley et al. have accentuated the need for caution when relying on self-reported symptoms as evidence of brain damage in patients involved in litigation, since they are more likely to endorse post-concussion symptoms [34]. Additionally, response bias might also be portrayed in this study. Patients who did not complete the RPQ might be less likely to partake in the follow-up than patients who did experience symptoms [26]. There was no detailed information on pre-morbid personality traits [35,36] and limited on the psychological distress of patients [37]. These factors could all potentially influence the reported outcome after mTBI. Furthermore, the RPQ was collected through various ways, and the method of administration could have influenced patients’ symptom reporting [23,38]. Lastly, there is a broad spectrum of abnormalities within the complicated mTBI group and doctors might treat patients differently when objective evidence for the brain injury was found [21]. Moreover, the confirmation of structural damage to the brain provided by imaging studies showing traumatic abnormalities (e.g., complicated mTBI) might lead to a higher rate of self-reported symptoms. 

For future research it would be recommended to look at the localization of abnormalities in CT or MRI data and see how this may affect post-concussion symptom reporting since in previous research CT abnormalities have been found to be related to outcome [39]. Additionally, brain imaging methods and technology are advancing, and this could potentially help to revolutionize and improve our understanding and detection of small changes in the brain following mTBI [3,24]. Furthermore, looking into the differences in treatment and treatment policies between complicated and uncomplicated mTBI would establish a better understanding considering the outcome. It would also be interesting to determine the occurrence of post-concussion symptoms and PCS at one year or even later follow-up times. Lastly, the DSM-V edition did not include PCS, but introduced the term mild neurocognitive impairment (MNI) due to TBI instead, which shows there is a move away from using PCS in mTBI research.

## 5. Conclusions 

This study showed that patients after complicated mTBI reported more post-concussion symptoms and have higher PCS prevalence rates compared to patients with uncomplicated mTBI at three and six months, which presents complicated mTBI as an indicator for these problems. However, the differences between both patient groups are small, and after adjusting for baseline covariates, this association could be explained by differences in baseline characteristics. These findings highlight the need to take the long-term impact on outcome for patients diagnosed with mTBI into consideration, and both patient groups are in need of clinical follow-up.

## Figures and Tables

**Figure 1 jcm-08-01921-f001:**
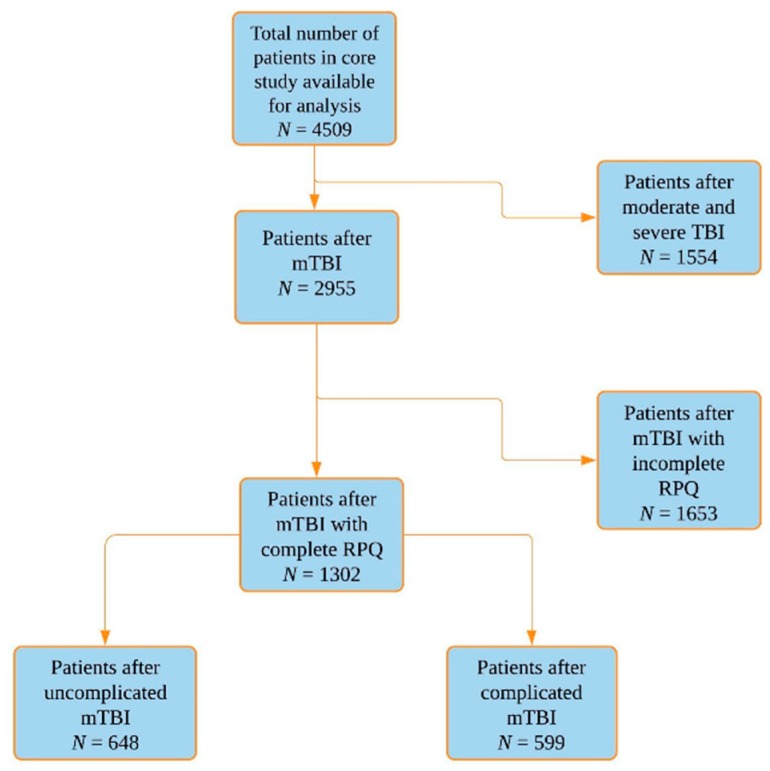
Flowchart sample size. *N*, number; mTBI, mild traumatic brain injury; RPO, Rivermead Post-Concussion Symptoms Questionnaire.

**Figure 2 jcm-08-01921-f002:**
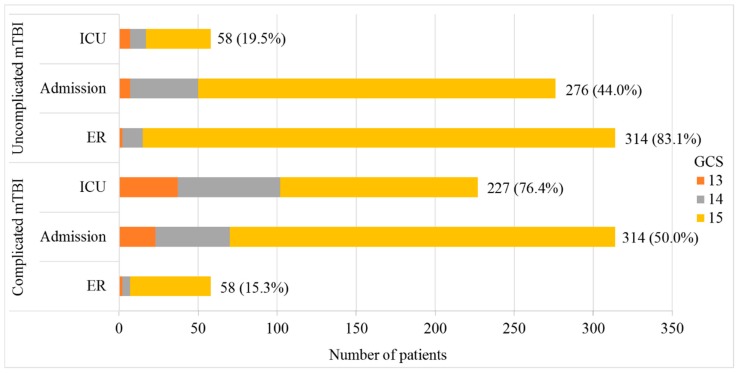
Number of uncomplicated and complicated mTBI patients with complete Rivermead Post-Concussion Symptoms Questionnaire (RPQ) data per GCS level 13–15 per stratum. mTBI, mild traumatic brain injury; GCS, Glasgow Coma Score; ICU, intensive care unit; ER, emergency room. *n* = 1302, complete case analysis.

**Figure 3 jcm-08-01921-f003:**
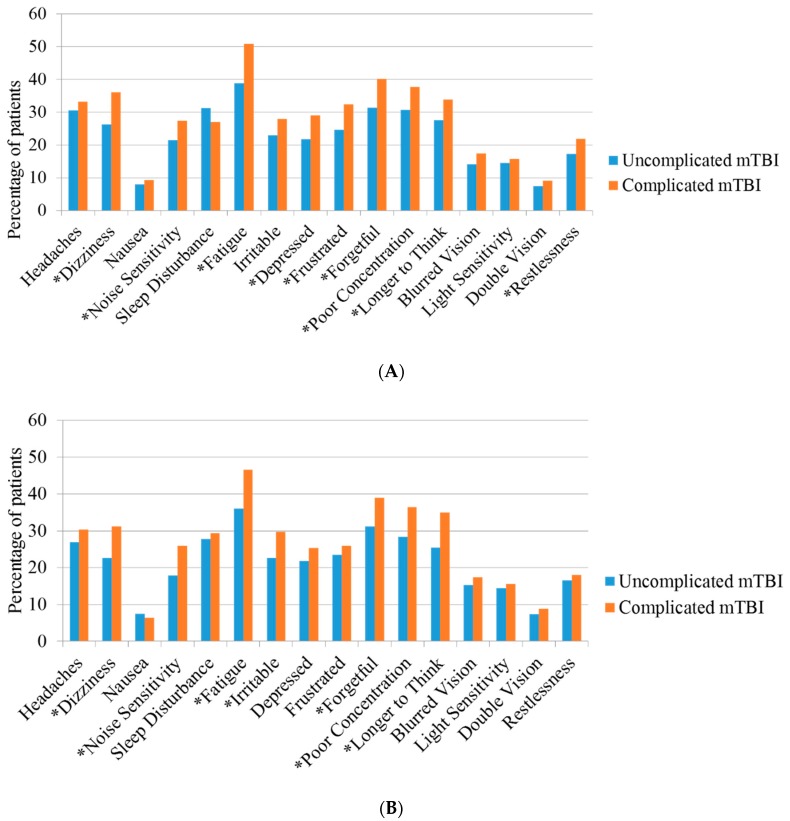
(**A**) Frequency of post-concussion symptoms with a severity rating of two (mild problem) or higher at 3 months. (**B**). Frequency of post-concussion symptoms with a severity rating of two (mild problem) or higher at 6 months. * significant (*p* < 0.05).

**Figure 4 jcm-08-01921-f004:**
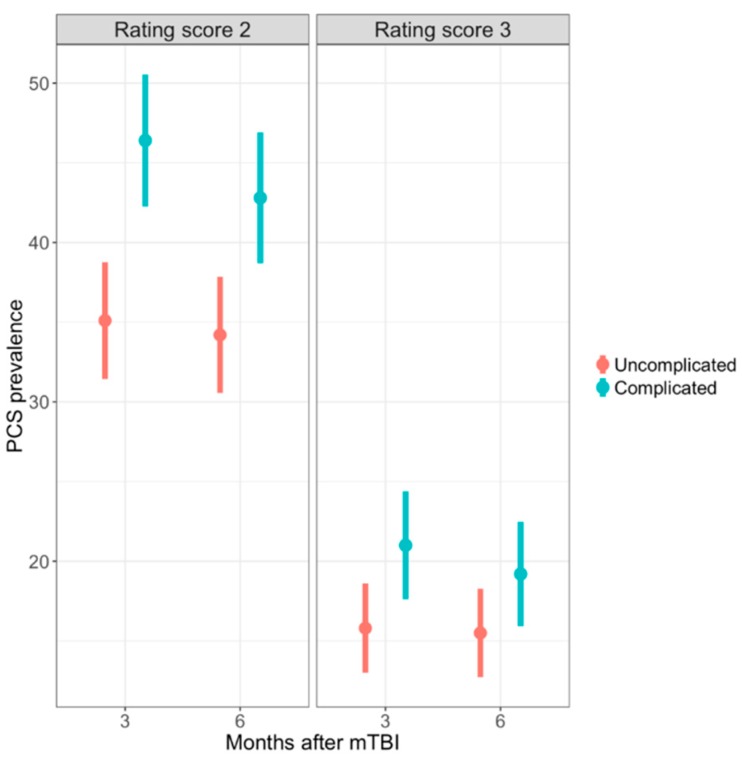
Prevalence of PCS for patients after uncomplicated and complicated mTBI at 3 and 6 months. PCS, post-concussion syndrome; mTBI, mild traumatic brain injury. Rating score of two = mild or worse; Rating score of three = moderate or worse.

**Figure 5 jcm-08-01921-f005:**
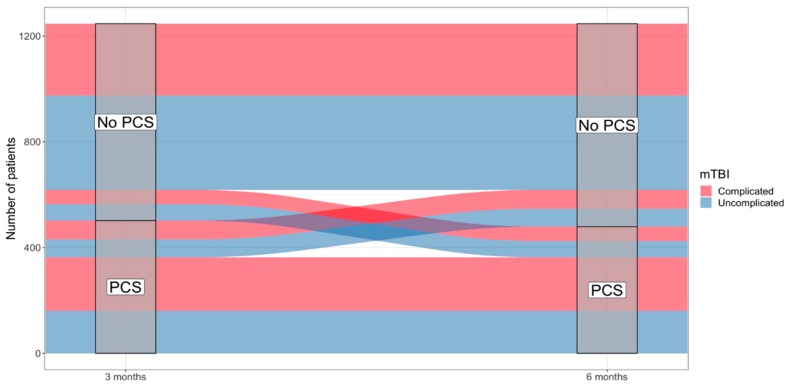
Trajectories of patients after uncomplicated and complicated mTBI with PCS at 3 and 6 months follow-up. PCS, post-concussion syndrome; mTBI, mild traumatic brain injury; *n* = number of patients.

**Figure 6 jcm-08-01921-f006:**
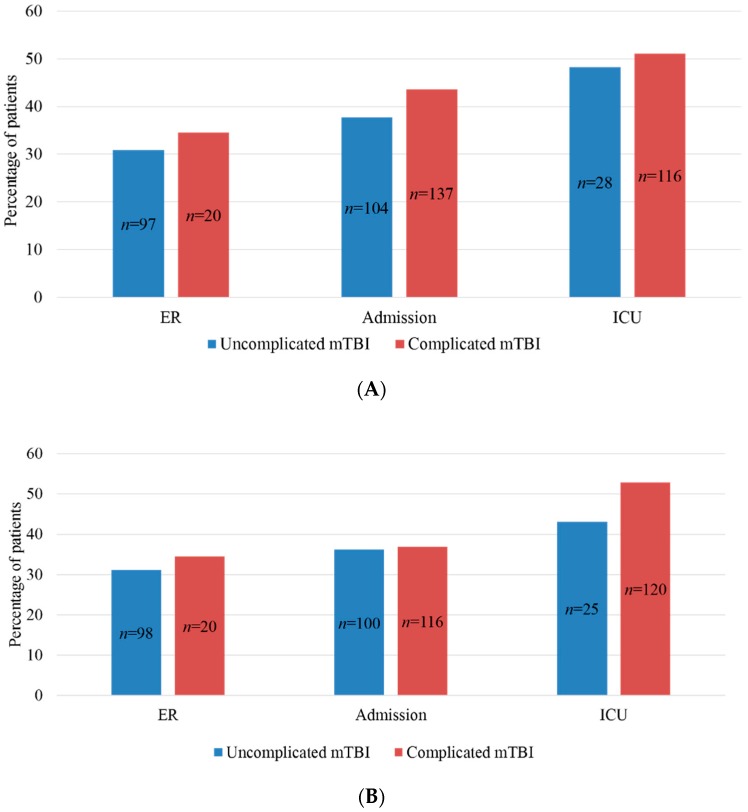
(**A**) Number and percentage of uncomplicated and complicated mTBI with PCS differentiated by strata at 3 months. (**B**) Number and percentage of uncomplicated and complicated mTBI with PCS differentiated by strata at 6 months. PCS, post-concussion syndrome; mTBI, mild traumatic brain injury; ER, emergency room; ICU, intensive care unit.

**Table 1 jcm-08-01921-t001:** Number of mild traumatic brain injuries (mTBIs), uncomplicated, and complicated mTBI per stratum.

	Total mTBI	Uncomplicated mTBI	Complicated mTBI
ER *N* (%)	826 (97.4%)	699 (84.6%)	97 (11.7%)
Admission *N* (%)	1409 (92.5%)	686 (48.7%)	627 (44.5%)
ICU *N* (%)	720 (33.7%)	144 (20.0%)	527 (73.2%)
Total	2955 (65.5%)	1529 (51.7%)	1251 (42.3%)

mTBI, mild traumatic brain injury; ER, emergency room; ICU, intensive care unit. Note: for 175 mTBI patients the CT scan was not available.

**Table 2 jcm-08-01921-t002:** Characteristics of the study population.

	Total mTBI with Completed RPQ	Uncomplicated	Complicated	*p*-Value
*N*	1302	648	599	
Gender (male)	827 (63.5%)	398 (61.4%)	396 (66.1%)	0.085
Age ^1^ (years)	53 (35–66)	51 (31.25–64)	58 (39–68)	<0.01
Education ^1^ (years)	14 (11–17)	14 (12–17)	13 (11–17)	0.054
Injury Mechanism				0.394
Road traffic accident	504 (38.7%)	255 (39.4%)	227 (37.9%)	
Incidental fall	616 (47.3%)	300 (46.3%)	289 (48.2%)	
Other non-intentional	72 (5.5%)	41 (6.3%)	29 (4.8%)	
Violence/assault	43 (3.3%)	22 (3.4%)	19 (3.2%)	
Act of mass violence	1 (0.1%)		1 (0.2%)	
Suicide attempt	7 (0.5%)	2 (0.3%)	5 (0.8%)	
Other	42 (3.2%)	23 (3.5%)	18 (3.0%)	
Unknown	17 (1.3%)	5 (0.8%)	11 (1.8%)	
Psychiatric Medical History	146 (11.2%)	68 (10.5%)	73 (12.2%)	0.329
Anxiety	36 (24.7%)	13 (19.1%)	22 (30.1%)	0.130
Depression	89 (61.0%)	46 (67.6%)	41 (56.2%)	0.161
Sleep disorders	19 (13.0%)	8 (11.8%)	10 (13.7%)	0.731
Schizophrenia	3 (2.1%)	1 (1.5%)	2 (2.7%)	0.602
Substance abuse disorder	17 (11.6%)	5 (7.4%)	9 (12.3%)	0.324
Other	19 (13.0%)	9 (13.2%)	9 (12.3%)	0.872
GCS baseline ^1^	15 (15–15)	15 (15–15)	15 (14–15)	<0.01
Computed Tomography				
Any intracranial injury on first CT	599 (46.0%)	648 (0.0%)	599 (100.0%)	<0.01
RPQ total score ^1^				
3 months	6 (0–17)	4 (0–14)	7 (2–20)	<0.01
6 months	5 (0–15)	4 (0–14)	7 (2–17)	<0.01

^1^ Data are displayed as median, with the first and third quartile given within brackets. mTBI, mild traumatic brain injury; RPQ, Rivermead Post-Concussion Symptoms Questionnaire; CT, Computed Tomography. Note: only completed RPQ (all items of the questionnaire filled out).

**Table 3 jcm-08-01921-t003:** Summary of covariate adjusted analysis for the association between complicated versus uncomplicated mTBI on the presence of PCS at 3 months and 6 months.

	3 Months		6 Months	
	OR	95% CI	OR	95% CI
Unadjusted	1.54	1.22–1.94	1.39	1.10–1.76
Adjusted	1.25	0.95–1.66	1.07	0.80–1.42

Baseline covariates adjusted for: age, gender, education, injury mechanism, GCS, complicated vs. uncomplicated, psychiatric medical history and stratum. mTBI, mild traumatic brain injury; PCS, post-concussion syndrome; OR, Odds Ratio; 95% CI, 95% confidence interval.

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
