# Peer review of "Post-Concussion Symptoms in Complicated vs. Uncomplicated Mild Traumatic Brain Injury Patients at Three and Six Months Post-Injury: Results from the CENTER-TBI Study"

_jcm, 2019, doi:10.3390/jcm8111921_

Round 1

Reviewer 1 Report

Post-concussion symptoms in complicated vs uncomplicated mild traumatic brain injury patients at three and six months post-injury: Results from the CENTER-TBI study.

Overall this is an interesting study with a large sample of people with mild traumatic brain injury (mTBI) with findings that are likely to be of interest to mTBI researchers and clinicians.  The list of authors is impressive.

My main issues with the study relate to inclusion of post-concussion syndrome (PCS) in the analyses, reliance on self-reported symptoms without presenting objective indications of outcome, selection bias, absence of control for psychological factors known to influence outcomes after mTBI as well as no mention of potential for repeated injury, given worsening of symptoms over time for some participants.

Briefly each in turn:

1. PCS analyses

I think it is a weakness of the study to include PCS analyses and I am unclear what these add.  There is no mention of more current diagnostic criteria sets (DSM-V and ICD-II) that have eschewed the PCS as a diagnostic category given the significant concerns and disagreements regarding the nature, etiology, and classification of the PCS.  The authors justify use of older diagnostic criteria sets because “the concept [PCS] is still used in the majority of post concussion symptom research.”  While possibly true, there is a move away from using PCS in mTBI research. The paper would be more focused if the analyses centred around post concussion symptoms rather than including both post concussion symptoms and PCS.

2. Self-reported symptoms

Reliance on self-report is a limitation not just because being involved in litigation wasn’t captured. Self-report symptom checklists query subjective symptoms that are not necessarily specific to mTBI, and which may over-estimate poor outcomes, especially as time passes after injury. Symptom scores generally do not correlate well with objective measures and can be influenced by situational factors.

3. Selection bias

How many questionnaires were sent out and how many face to face interviews were completed? There are known differences in responses to self-report questionnaires based on method of administration.

Further, information about the sample size is confusing. There were three different sample sizes reported: n = 1,718 (p 3, line 128), n = 2,955 (page 5, Table 1), n = 1,302 (page 5, line 201).  I found it difficult to decipher what the actual sample was for this study.  How many participants with data at baseline (ie who met criteria for inclusion in the study) were lost to follow up over time. Is there any information about this group, were there differences in loss to follow up between the complicated and uncomplicated groups or by stratum?

4. Absence of information about the impact of secondary psychological factors is disappointing given how significant such factors are in understanding outcomes after mTBI, especially over time.

5. Do the authors have any information about the 54 people who reported more symptoms at 6 months than 3 months? For example was any information collected about repeated injuries?  Secondary psychological factors may also contribute to this trajectory.

6. Given all these concerns and risks for bias I think the findings of the study have been over-stated in the conclusion section on page 13.

Other more minor issues:

For included patients with a GCS score of 15, were other markers available to confirm the diagnosis of mTBI?

What were the reasons for ICU admissions?  Were these admissions mTBI-related or related to other injuries sustained at the same time or non-mTBI related complications?

Are the reported RPQ median scores in Table 2 correct?  These scores seem extremely low. Were these just scores for the seven symptoms selected to map to the ICD-10 PCS diagnosis? If these really are the scores for all 16 items then Id question clinical significance.

Author Response

Reviewer: 1

Comments and Suggestions for Authors

Overall this is an interesting study with a large sample of people with mild traumatic brain injury (mTBI) with findings that are likely to be of interest to mTBI researchers and clinicians.  The list of authors is impressive.

My main issues with the study relate to inclusion of post-concussion syndrome (PCS) in the analyses, reliance on self-reported symptoms without presenting objective indications of outcome, selection bias, absence of control for psychological factors known to influence outcomes after mTBI as well as no mention of potential for repeated injury, given worsening of symptoms over time for some participants.

Briefly each in turn:

Comment #1

PCS analyses

I think it is a weakness of the study to include PCS analyses and I am unclear what these add.  There is no mention of more current diagnostic criteria sets (DSM-V and ICD-II) that have eschewed the PCS as a diagnostic category given the significant concerns and disagreements regarding the nature, etiology, and classification of the PCS.  The authors justify use of older diagnostic criteria sets because “the concept [PCS] is still used in the majority of post concussion symptom research.”  While possibly true, there is a move away from using PCS in mTBI research. The paper would be more focused if the analyses centred around post concussion symptoms rather than including both post concussion symptoms and PCS.

Author’s response #1

We would like to thank the reviewer for these thoughtful comments.

The main focus of the paper is to show the occurrence of post-concussion symptoms  after complicated and uncomplicated mTBI at three and six months post-injury, and see if there are differences between these two groups. We fully agree that the use of the term “PCS syndrome” is controversial – as stated in the introduction. In this manuscript, we refer to PCS syndrome as being characterized by 3 or more PCS symptoms according to the ICD-10 criteria. We do not wish to imply that this combination in fact represents a syndrome. We thought of using a possible different terminology like “more severe PCS symptoms”, but decided against this (using rating score 2 as a cut-off for example may not really be qualified as severe) and prefer to retain the phrasing syndrome, also for comparability with other studies and to make it more generalizable to previous research done in this field

Additionally, we first looked at the frequency of post-concussion symptoms, and then we focused on the syndrome. The symptoms are the underlying structure for the syndrome. Patterns on symptom and syndrome level describe similarities, therefore it is unlikely that the message of the paper would be very different when the paper would be more centered around post-concussion symptoms. Figure 3.1 and 3.2, and Appendix C.1 and C.2 are the backbone of the PCS figures.

We have adopted a conventional definition of PCS, however, we are very aware that the PCS has been dropped from the DSM-V. For this reason, we also slightly revised the discussion paragraph, to show that we are aware of the concerns considering PCS as a diagnostic category, and that there is no mention in the DSM-V[1].

“Lastly the DSM-V edition did not include PCS, but introduced the term mild neurocognitive impairment [MNI] due to TBI instead, which shows there is a move away from using PCS in mTBI research.”

Lastly, objective assessments are essential, but the DSM-5 does not name any proprietary tests. The DSM-5 also states that the neurocognitive symptoms following mTBI “tend to resolve within days to weeks after the injury with complete resolution typical by 3 months”, which has been shown in previous research (and is also mentioned in our introduction), to not be the case for a certain percentage of patients.

Comment #2

Self-reported symptoms

Reliance on self-report is a limitation not just because being involved in litigation wasn’t captured. Self-report symptom checklists query subjective symptoms that are not necessarily specific to mTBI, and which may over-estimate poor outcomes, especially as time passes after injury. Symptom scores generally do not correlate well with objective measures and can be influenced by situational factors.

Author’s response #2

We agree with the reviewer that being involved in a litigation is not the only reason why relying on self-report is a limitation.

The RPQ is based on self-report and subjects are asked to rate their symptom severity “on the day prior to completing the questionnaire”. In addition, the diagnosis of PCS permits that “symptoms cause clinical significant impairment”. This is also not measured with the RPQ. Therefore we added the following sentences in our methods section:

“It must be emphasized that the RPQ cannot accurately diagnose PCS, since it is based on self-report rather than clinical examination and the symptoms assessed may not be specific to mTBI. Additionally, it does not include information on the duration of the symptoms and clinical significant impairment[2].”

Comment #3

Selection bias

How many questionnaires were sent out and how many face to face interviews were completed? There are known differences in responses to self-report questionnaires based on method of administration.

Further, information about the sample size is confusing. There were three different sample sizes reported: n = 1,718 (p 3, line 128), n = 2,955 (page 5, Table 1), n = 1,302 (page 5, line 201).  I found it difficult to decipher what the actual sample was for this study.  How many participants with data at baseline (ie who met criteria for inclusion in the study) were lost to follow up over time. Is there any information about this group, were there differences in loss to follow up between the complicated and uncomplicated groups or by stratum?

Author’s response #3

We thank the reviewer for bringing up confusion concerning the method of administration and the sample size. We therefore have added a table in the supplementary material (Appendix A) in which we clarify the methods of administration for the RPQ at 3 and 6 months.

We also added the following sentence as a limitation to the discussion section:

“Additionally, response bias might also be portrayed in this study. Patients who did not complete the RPQ might be less likely to partake in the follow up than patients who did experience symptoms[3].

Furthermore, we have generated a flowchart (Figure 1) in which we explain the sample and show how many participants were lost to follow up over time. In the text we already mentioned the following: “Patients after mTBI who completed the RPQ were not significantly different from those with incomplete RPQ data, except that they had a slightly higher number of education years (p<0.01: 13.9 vs. 12.6) and more patients reported to have had a psychiatric medical history (p<0.01). Additionally, there was no statistically significant difference between patients who had a completed RPQ at three and/or six months (p=0.17)”.

We have added a table in the supplementary material (Appendix B) in which the differences between patients with a completed and uncompleted RPQ are portrayed.

Comment #4

Absence of information about the impact of secondary psychological factors is disappointing given how significant such factors are in understanding outcomes after mTBI, especially over time.

Author’s response #4

We also agree with the reviewer that the impact of secondary psychological factors is something that should not be overlooked. For this reason, we have added psychiatric medical history variables, such as anxiety, depression, sleep disorders, schizophrenia, substance abuse disorder, into our analyses. When comparing the models with and without these variables, we do see a slight added value (R2 from 0.061 to 0.072).

In the methods section we have adjusted the number from N=1,718 to N=1,302 (line 128). This was to eliminate some of the confusion regarding the numbers used for data analysis.

Comment #5

Do the authors have any information about the 54 people who reported more symptoms at 6 months than 3 months? For example was any information collected about repeated injuries?  Secondary psychological factors may also contribute to this trajectory.

Author’s response #5

Unfortunately, specific information about repeated injuries was not collected within the CENTER-TBI study. We did capture specific information on general health and other inter current conditions on follow-up. This would also include a possible 2nd TBI. However, detailed information on a possible 2nd TBI was not captured. We do have information available if people have had a TBI before they were included in the study, however, this would not explain the shift from not meeting the classification criteria at 3 months but meeting them at 6 months.

Comment #6

Given all these concerns and risks for bias I think the findings of the study have been over-stated in the conclusion section on page 13.

Author’s response #6

As the reviewer suggests that the findings of the study have been over-stated, we have changed the conclusion a bit:

“This study showed that patients after complicated mTBI reported more post-concussion symptoms and have higher PCS prevalence rates compared to patients with uncomplicated mTBI at three and six months, which presents complicated mTBI as an indicator for these problems. However, the differences between both patient groups are small, and after adjusting for baseline covariates, this association could be explained by differences in baseline characteristics. These findings highlight the need to take the long-term impact on outcome for patients diagnosed with mTBI into consideration, and both patient groups are in need of clinical follow-up.”

Other more minor issues:

Comment #7

For included patients with a GCS score of 15, were other markers available to confirm the diagnosis of mTBI?

Author’s response #7

Comment #8

What were the reasons for ICU admissions?  Were these admissions mTBI-related or related to other injuries sustained at the same time or non-mTBI related complications?

Author’s response #8

The reviewer raises a very good point and this is something that has been looked at extensively within the CENTER-TBI study. In a recently published manuscript reporting the main descriptive results of CENTER-TBI, we stated:

“Patients with mild TBI (GCS >12) constituted a third of patients in the ICU stratum. Plausible explanations for these ICU admissions include advanced age, frailty, comorbidities, increased risks of lesion progression due to use of anticoagulants and antiplatelet drugs, and the need for (extracranial) surgery[4].”
“The stratification of patients by care pathway showed clear discordances with the GCS-based classification of TBI severity[4].”

Additionally, there are also substantial difference between countries in pre-hospital care and treatment policies, which was shown by the provider profiling questionnaires, and the above paper.

Comment #9

Are the reported RPQ median scores in Table 2 correct?  These scores seem extremely low. Were these just scores for the seven symptoms selected to map to the ICD-10 PCS diagnosis? If these really are the scores for all 16 items then Id question clinical significance.

Author’s response #9

We have double checked the scores once more, and the reported median scores in Table 2  are indeed correct.     The data is heavily skewed to zero, because approximately 1/3rd  of the patients do not experience any symptoms (3 months: 29.5%, 6 months:   30.6%)

This is based on the total score of all 16 items (max 64: 16 symptoms * 4), and not on the seven selected symptoms. These numbers are similar to numbers that have been reported in previous research [3].

References

American Psychiatric Association. Diagnostic and statistical manual of mental disorders, 5th ed. (dsm-5). 2013. Cnossen, M.C.; Winkler, E.A.; Yue, J.K.; Okonkwo, D.O.; Valadka, A.; Steyerberg, E.W.; Lingsma, H.; Manley, G.T.M.D.P.D. Development of a prediction model for post-concussive symptoms following mild traumatic brain injury: A track-tbi pilot study. J Neurotrauma 2017. Voormolen, D.C.; Cnossen, M.C.; Polinder, S.; von Steinbuechel, N.; Vos, P.E.; Haagsma, J.A. Divergent classification methods of post-concussion syndrome after mild traumatic brain injury: Prevalence rates, risk factors, and functional outcome. J Neurotrauma 2018, 35, 1233-1241. Steyerberg, E.; Wiegers, E.; Sewalt, C.; Maas, A.I. Case-mix, care pathways, and outcomes in patients with traumatic brain injury in center-tbi: A european prospective, multicentre, longitudinal, cohort study. The Lancet Neurology 2019, In press.

Reviewer 2 Report

The study by Voormolen and colleagues examined the occurrence of post-concussive symptoms and PCS in a cohort drawn from the multinational CENTER-TBI study. By further dividing participants into complicated and uncomplicated mTBI based on presence or absence of abnormal CT findings, the authors aimed to determine the association between complicated mTBI and PCS. The authors have performed a thorough investigation which adds to the ongoing elucidation of complicated mTBI.

Major comments:

Line 128: study N reported as 1,718, but total mTBI with completed RPQ is reported in table 2 as N = 1,302. Methods state that authors included only subjects with CT findings and RPQ at three and six months in order to perform a complete case analysis, so the number of participants meeting this criteria should be described more clearly as it was confusing for the reader. Since the main study has been reported elsewhere and this study focuses on a subgroup, it would be best to remove table 1 and begin results at line 208, where this population is described.  The authors make an important point regarding medical treatment of complicated patients on line 316 (discussion). The increased rate of medical attention/treatment and investigation complicated patients typically receive may skew their self-reported symptoms. If possible, authors should expand on this in the context of their findings. "Enriched population" may not be clear to all readers - it would help to describe enrichment as it pertains to clinical trials in the introduction.

Minor comments: 

Sentence beginning line 80 requires revising - perhaps missing a word? Sentence beginning line 296 should be expanded on - what is the conundrum? Figure 4, figure 5.1 & 5.2 - y axis labels are missing.

Author Response

Reviewer: 2

Comments and Suggestions for Authors

The study by Voormolen and colleagues examined the occurrence of post-concussive symptoms and PCS in a cohort drawn from the multinational CENTER-TBI study. By further dividing participants into complicated and uncomplicated mTBI based on presence or absence of abnormal CT findings, the authors aimed to determine the association between complicated mTBI and PCS. The authors have performed a thorough investigation which adds to the ongoing elucidation of complicated mTBI.

Major comments:

Comment #1

Line 128: study N reported as 1,718, but total mTBI with completed RPQ is reported in table 2 as N = 1,302. Methods state that authors included only subjects with CT findings and RPQ at three and six months in order to perform a complete case analysis, so the number of participants meeting this criteria should be described more clearly as it was confusing for the reader. Since the main study has been reported elsewhere and this study focuses on a subgroup, it would be best to remove table 1 and begin results at line 208, where this population is described. 

Author’s response #10

We thank the reviewer for bringing up this point, and since it was suggested by both reviewers, we have made changes in the text accordingly.

Furthermore, we have generated a flowchart (Figure 1) in which we explain the sample and show how many participants were lost to follow up over time, which should provide the clarification asked for.

We have not deleted table 1, since we believe the flowchart provides detailed information and takes away the previous confusion.

Comment #11

The authors make an important point regarding medical treatment of complicated patients on line 316 (discussion). The increased rate of medical attention/treatment and investigation complicated patients typically receive may skew their self-reported symptoms. If possible, authors should expand on this in the context of their findings. "Enriched population" may not be clear to all readers - it would help to describe enrichment as it pertains to clinical trials in the introduction.

Author’s response #11

We have expanded on the sentence at the end of the limitations section:

“Moreover, the confirmation of structural damage to the brain provided by imaging studies showing traumatic abnormalities (e.g. complicated mTBI) might lead to a higher rate of self-reported symptoms”

And we briefly pointed towards enrichment in the introduction by adding the following sentence at the end of the hypotheses paragraph:

“This would be particular relevant when planning a clinical trial investigating efficacy of approaches to treat PCS symptoms”

Lastly, we swapped the hypothesis paragraph and objectives in the introduction to make the storyline flow better.

Minor comments:

Comment #2

Sentence beginning line 80 requires revising - perhaps missing a word?

Author’s response #2

We have changed the sentence into:

“A certain percentage of patients (estimated between 5–43%[1-6]) after mTBI report and experience post-concussion symptoms for months and sometimes even longer post-injury [7,8].”

Comment #3

Sentence beginning line 296 should be expanded on - what is the conundrum?

Author’s response #3

The following sentence has been added for clarification about the conundrum:

“which is based on the question if patients after complicated mTBI are similar or dissimilar based on symptom reporting compared to uncomplicated mTBI patients.”

Comment #4

Figure 4, figure 5.1 & 5.2 - y axis labels are missing.

Author’s response #4

We have added y-axis labels  to figure 6.1 and 6.2 (previous version 5.1 and 5.2) and have made the label clearer in Figure 5 (previous version 5).

References

Binder, L.M. Persisting symptoms after mild head injury: A review of the postconcussive syndrome. J Clin Exp Neuropsychol 1986, 8, 323-346. Leddy, J.J.; Sandhu, H.; Sodhi, V.; Baker, J.G.; Willer, B. Rehabilitation of concussion and post-concussion syndrome. Sports Health 2012, 4, 147-154. Spinos, P.; Sakellaropoulos, G.; Georgiopoulos, M.; Stavridi, K.; Apostolopoulou, K.; Ellul, J.; Constantoyannis, C. Postconcussion syndrome after mild traumatic brain injury in western greece. J Trauma 2010, 69, 789-794. King, N.S.; Kirwilliam, S. The nature of permanent post-concussion symptoms after mild traumatic brain injury. Brain Impairment 2013, 14, 235-242. Ruff, R.M. Mild traumatic brain injury and neural recovery: Rethinking the debate. NeuroRehabilitation 2011, 28, 167-180. Hiploylee, C.; Dufort, P.A.; Davis, H.S.; Wennberg, R.A.; Tartaglia, M.C.; Mikulis, D.; Hazrati, L.N.; Tator, C.H. Longitudinal study of postconcussion syndrome: Not everyone recovers. J Neurotrauma 2016. Dikmen, S.; Machamer, J.; Temkin, N. Mild traumatic brain injury: Longitudinal study of cognition, functional status, and post-traumatic symptoms. J Neurotrauma 2017, 34, 1524-1530. Smits, M.; Hunink, M.G.M.; van Rijssel, D.A.; Dekker, H.M.; Vos, P.E.; Kool, D.R.; Nederkoorn, P.J.; Hofman, P.A.M.; Twijnstra, A.; Tanghe, H.L.J., et al. Outcome after complicated minor head injury. American Journal of Neuroradiology 2008, 29, 506.

Round 2

Reviewer 1 Report

The authors have adequately addressed my original concerns. The addition of the flow chart clarifying the sample size is especially helpful.